# Self-Medication as an Important Risk Factor for Antibiotic Resistance: A Multi-Institutional Survey among Students

**DOI:** 10.3390/antibiotics11070842

**Published:** 2022-06-23

**Authors:** Shah Zeb, Mariam Mushtaq, Muneeb Ahmad, Waqas Saleem, Ali A. Rabaan, Bibi Salma Zahid Naqvi, Mohammed Garout, Mohammed Aljeldah, Basim R. Al Shammari, Nehad J. Al Faraj, Nisreen A. Al-Zaki, Mona J. Al Marshood, Thuria Y. Al Saffar, Khadija A. Alsultan, Shamsah H. Al-Ahmed, Jeehan H. Alestad, Muhammad Naveed, Naveed Ahmed

**Affiliations:** 1Department of Microbiology, Faculty of Biomedical and Health Science, The University of Haripur, Haripur 22610, Pakistan; microbiologist018@gmail.com (S.Z.); szm.pro@gmail.com (B.S.Z.N.); 2Department of Microbiology, Pakistan Kidney and Liver Institute & Research Center, Lahore 54000, Pakistan; waqas.saleem@pkli.org.pk; 3Department of Medical Education, King Edward Medical University, Lahore 54000, Pakistan; mariammushtaq105@gmail.com; 4Department of Medical Education, Rawalpindi Medical University, Rawalpindi 46000, Pakistan; muneeb316161@gmail.com; 5Molecular Diagnostic Laboratory, Johns Hopkins Aramco Healthcare, Dhahran 31311, Saudi Arabia; 6College of Medicine, Alfaisal University, Riyadh 11533, Saudi Arabia; 7Department of Public Health and Nutrition, The University of Haripur, Haripur 22610, Pakistan; 8Department of Community Medicine and Health Care for Pilgrims, Faculty of Medicine, Umm Al-Qura University, Makkah 21955, Saudi Arabia; magarout@uqu.edu.sa; 9Department of Clinical Laboratory Sciences, College of Applied Medical Sciences, University of Hafr Al Batin, Hafr Al Batin 39831, Saudi Arabia; mmaljeldah@uhb.edu.sa (M.A.); bralshammari@uhb.edu.sa (B.R.A.S.); 10Specialty Paediatric Medicine, Qatif Central Hospital, Qatif 32654, Saudi Arabia; farajnj@hotmail.com (N.J.A.F.); nisreen.alzaki@yahoo.com (N.A.A.-Z.); drmjmm@gmail.com (M.J.A.M.); talsafar@moh.gov.sa (T.Y.A.S.); drkmohsen@hotmail.com (K.A.A.); drshamsahmed123@gmail.com (S.H.A.-A.); 11Immunology and Infectious Microbiology Department, University of Glasgow, Glasgow G1 1XQ, UK; jeehanalostad@gmail.com; 12Microbiology Department, Collage of Medicine, Jabriya 46300, Kuwait; 13Department of Biotechnology, Faculty of Life Sciences, University of Central Punjab, Lahore 54000, Pakistan; 14Department of Medical Microbiology and Parasitology, School of Medical Sciences, Universiti Sains Malaysia, Kubang Kerian 16150, Kelantan, Malaysia

**Keywords:** antimicrobial stewardship, Pakistan, antibiotic knowledge, health belief model, self-medication

## Abstract

Self-medication is an important issue, especially in developing countries. Self-medication is the concept in which individuals use medicine to ease and manage their minor illnesses. The current survey was designed to conduct interviews at different universities based on the availability of the students from August 2021 to October 2021 in Hazara region of Khyber Pakhtunkhwa (KPK), Pakistan. Overall, 1250 questionnaires were distributed to students from various departments. Students of microbiology (*n* = 305, 24.4%) and agriculture 236 (*n* = 18.8%) were the most elevated members in this study, while other participants were from medical lab technology (*n* = 118, 9.4%), chemistry (*n* = 103, 8.2%), food science (*n* = 92, 7.3%), business administration (*n* = 83, 6.6%), sociology (*n* = 78, 6.2%), math/physics (*n* = 6, 14.8%), Pak study (*n* = 58, 4.6%), English (*n* = 47, 3.7%), and psychology (*n* = 19, 1.5%). Students working towards their Bachelor numbered (*n* = 913, 73.0%), Master (minor) numbered (*n* = 80, 6.4%), Master (major) numbered (*n* = 221, 17.6%), and Doctorate numbered (*n* = 36, 2.8%). The age group of participants was majorly 20–25 years (61.0%), while others belonged to the age groups 25–30 years (20.6%), 30–35 years (9.8%), and 35–40 years (8.4%). The mean and standard deviation of daily practices of self-medication were observed (M = 416.667, SD = 1,026,108.667) and *p* = 0.002. The mean and standard deviation of daily practices of antibiotic knowledge was (M = 431.5, SD = 1,615,917) and *p* = 0.002. Antimicrobial agents were leading over others with 631 (50.4%), followed by anti-inflammatory with 331 (26.4%), multivitamins with 142 (11.3%), gynecological purpose with 59 (4.7%), and analgesic with 72 (5.7%), while the lowest frequency rate was observed against herbal remedies with 15 (1.2%). The results of the current study concluded that students practiced self-medication for reasons such as convenience to obtain these medications from cheap sources and to avoid the fee of a physician. They searched for the medicine on social media platforms and purchased it blindly from the pharmacy without any prescription from a physician.

## 1. Introduction

Self-medication is defined as obtaining and consuming medicines without the advice of a physician for either diagnosis or surveillance of treatment. In other terms, self-medication is defined as the use of any drug not prescribed by a licensed health practitioner or the use of other non-prescription medicine [1,2]. The selection of a drug in order to treat an illness or symptom that is self-diagnosed is termed self-medication. Delay in seeking medical advice when required and instead of trying to heal oneself, masking a severe disease, has been one of the greatest risks of self-medication that may worsen a health situation. Responsible self-medication may be a proposed strategy for self-medication in cases of a common illness which a doctor has already diagnosed and prescribed some therapeutic products [3]. Another proposed way of self-medication considered safe by the World Health Organization is the provision of a leaflet with each medicine describing complete information about correct dosage, side effects, and duration. One cannot neglect herbal medicines, which are easily available over the counter and contribute largely towards self-medication but must also be maintained by apposite predetermined pros and cons of the medicine [4,5].

Self-medication is an important issue, especially in developing countries. Self-medication is a concept in which individuals use substances to ameliorate and manage their minor illnesses. Individuals are gambling with their health in the hope that some substances help their illness and the problem is better controlled [6,7]. It is one of the common and favored modes reestablished by the patient. Self-medication cannot be considered as totally harmful, as many drugs classified as over the counter can be purchased without a prescription and many times might save money and time for patients [8,9].

Self-medication among individuals with a healthcare background is thought to bring about independence, empowering them to make decisions to cope with minor health issues [10,11]. A number of benefits have been associated with self-medication for healthcare systems such as expediting a quick access to treatment with a reduced budget. Despite all the benefits, risks associated with self-medication cannot be overlooked. These risks may include excess dose of a particular drug, using the wrong drug in a given condition, prolonged usage of a certain drug, and taking different drugs at the same time that may cause a drug interaction [12,13]. Carmel et al., 2001, suggested a direct need of a proper monitoring system for the use of drugs and partnership between healthcare professionals, patients, and pharmacists, along with specific training and awareness programs about the risks of self-medication that may minimize these risks. Suggestions from friends, neighbors, families, and the pharmacist, or from an advertisement in the newspaper are common sources of self-medication [14]. The main problems related to self-medication are increased resistance to pathogens, causing serious health hazards such as prolonged suffering and adverse reactions. Nowadays, antimicrobial resistance is a worldwide problem that may be caused by antibiotics given without prescription [15,16,17].

Self-medication helps to reduce pressure on medical services where health care personnel are insufficient [18]. It plays an active role in a patient’s health care, self-dependence in preventing minor symptoms or conditions, and awareness and education opportunities on specific health problems. Products to treat heartburn and stop smoking, for example, offer a convenient and cheap source of medication. Self-medication increases the availability of health care to populations living in rural and remote areas [8]. Inaccurate self-diagnosis and the wrong choice of therapy may cause other complications. Failure to seek appropriate medical advice promptly, adverse effects and resistance to pathogens, and excessive or inadequate dosage are some potential risks of self-medication [19].

## 2. Results

### 2.1. Contributed Departments

A total of 1400 students responded the questionnaire survey on time. Responses with inadequate and rehashed answers were excluded from the study. After exclusion, a total of 1250 questionnaires were incorporated for research analysis. Most of the students were from the departments of microbiology (*n =* 305, 24.4%) and agriculture (*n =* 236, 18.8%), along with students from other departments such as medical lab technology (*n =* 118, 9.4%), chemistry (*n =* 103, 8.2%), food science (*n =* 92, 7.3%), business administration (*n =* 83, 6.6%), sociology (*n =* 78, 6.2%), math/physics (*n =* 61, 4.8%), Pak study (*n =* 58, 4.6%), English (*n =* 47, 3.7%), and psychology (*n =* 19, 1.5%). The variance (V) and standard deviation (SD) for departments/participants were calculated as V = 7140.97436 and SD = 84.50429, while probability was observed as *p* < 0.005.

### 2.2. Disciplines Contributed

The participants were accounted for in the survey as per their discipline, such as Bachelor (*n =* 913, 73.0%), Master (minor) (*n =* 80, 6.4%), Master (major) (*n =* 221, 17.6%), and Doctorate (*n =* 36, 2.8%) (Figure 1). The overall V and SD based on discipline was V = 124,870.25 and SD = 353.36985, while probability was observed as *p* = 0.007.

### 2.3. Age Interpretation

Most of the study participants belong to age group 20–25 years (61.0%), while 25–30 years were 20.6%, 30–35 years were 9.8%, and 35–40 were 8.4% (Figure 2). The age and number of participants showed a negative correlation with a *p* < 0.001, while the mean and SD of age/number of participants was M = 26.154 and SD = 1334.308.

### 2.4. Daily Practices and Approach of Self-Medication in Students

For Question 1 (Q1), against antibiotics practices, positive responses (*n* = 1005, 80.4%) were seen by most of the students who were enrolled in health sciences. Moderate knowledge (*n* = 205, 16.4%) was shown by the natural science students. In Q2, the higher utilization of a broad spectrum of antibiotics was noted to be around (*n* = 750, 60%) among the undergraduate (bachelors) students with different divisions, as compared to a narrow spectrum (*n* = 355, 28.4%). Other routine self-medication was (*n* = 145, 11.6%). In Q3, exceptionally certain outcomes were found, for example (*n* = 991, 79.2%) did not experience any adverse reaction during self-medication because they had used medication appropriately according to recommendations from doctors, while (*n* = 259, 20.7%) were negative who experienced unfavorable adverse reactions due to high dose of self-medicine. In Q4, highly negative outcomes *(n =* 980, 78.4%) were found due to lack of awareness and information of self-medication hazards, while positive results were *(n =* 208, 16.6%), perhaps because these students were currently affiliated with healthcare campaigns as well as health sciences. In Q5, we explored reasons for self-medication which included that they were reliable *(n =* 173, 13.8) due to the storage of medicine at hostels or home, exempt from physician fees *(n =* 241, 19.2%) due to fixed pocket money at the hostel, that the student was unable to walk the distance to the physician *(n =* 199, 15.9%), or that they simply preferred self-medication, i.e., for self-satisfaction *(n =* 637, 50.9%) (Table 1).

### 2.5. Current Challenges and Antibiotics Mindset/Counseling in Students

For Q1 in Table 2, an updated mindset was found in 627 (50.1%) participants related to antibiotics resistance; this may have been due to the fact that most of the students kept an eye on news and publications regarding healthcare. An unaware mindset was found in 570 (45.6%) study participants, and a neutral mindset was found among 53 (4.2%) as they from a purely social sciences background. In Q2, negative results were found in 865 (69.2%) participants; they were aware of the toxicity related to high consumption of antibiotics because they were registered under health sciences, but they still preferred antibiotics therapy due to a speedy recovery as compared to other medicines or therapy. A total of 138 (11.0%) were unaware of antibiotics’ toxicity due to a natural and social science background, and 243 (19.7%) were neutral. Q3 also established negative results. Most of the students (782 (62.5%)) were aware of antibiotics resistance as they were registered under the health sciences, food sciences, or agriculture discipline, while the students of math/physics, zoology, and botany were found to be slightly aware of antibiotics resistance, i.e., 152 (12.1%), as compared to business administration and English (316 (25.2%)). Q4 generated a negative finding as well; 805 (64.4%) said that their physician did not explain about antibiotics’ toxicity and dangers, while 421 (36.6%) were guided by their physician and dentists, and 24 (1.9%) did not go through any counseling.

### 2.6. Overall Relationship between Daily Practices and Knowledge of Antibiotics among Students

Table 1 represents the practices of self-medication among students in routine life. Although a positive correlation (r = 0.0311) was seen between daily practices and knowledge of antibiotics among students, the inter-relationship of variables was weak. The mean and standard deviation of daily practices for self-medication was observed as M = 416.667 and SD = 1,026,108.667, with *p* = 0.002.

Table 2 represents the awareness of antibiotics misuse and risk of AMR among students during routine life. A positive correlation (r = 0.0311) was seen, while the inter-relationship between the variables was weak. The mean and standard deviation of daily practices of the knowledge about misuse of antibiotics was observed as M = 431.5 and SD = 1,615,917, with *p* = 0.002.

### 2.7. Self-Medication Practices in Students to Treat Disease/Illness

The most frequently used medicinal agents against illness/disease were observed, with antimicrobial agents leading over others at 631 (50.4%), then anti-inflammatory with 331 (26.4%), multivitamins with 142 (11.3%), gynecological purposes with 59 (4.7%), and analgesic with 72 (5.7%), while the lowest frequency rate was observed against herbal remedies with 15 (1.2%).

### 2.8. Relationship between Self-Practices and Treatment of Disease in Students

Although technically a positive correlation was observed between self-medication and students, the relationship between the variables was weak (Figure 3), while the mean was M = 124.6, SD = 597.4, and r = 0.1533.

### 2.9. Prevalence of Infection/Disease Treated by Self-Medication in Students

The illnesses and diseases among students such as coughing (2.4%), headache (33.5%), dengue (16.8%), body aches (4.2%), chest pain (0.42%), stomach pain (1.4%), flu/fever (24.6%), diarrhea (0.96%), anemia (7.5%), and menstruation (7.6%) were eliminated (Figure 4). Most of students treated the seasonal illness/disease through self-medication, even if they dealt with major problems too.

## 3. Discussion

One of the most common wrong practices in Pakistan is self-medication or the misuse of antibiotics in which the people usually take the medication without any consultation from physicians or clinicians [15,17,20]. Mindful of the worst AMR situation in Pakistan, the current study was conducted with the concept of self-medication among youth, asking why students choose self-medication, why a number of students were taking antibiotics, where did they gain knowledge about self-medication or to treat the illness/disease, and how did they combat seasonal and routine health issues.

A study from Tanzania has shown that self-medication with antibiotics was found to be common in 57% of cases, with amoxicillin (32.08%) being the most commonly used drug [3]. The findings of the current study have seen a higher rate of self-medication among Bachelor’s students as it helps to reduce pressure on medical services where healthcare personnel are insufficient. They were active members of the social platform, had a good relationship with the healthcare professionals, and some were affiliated with healthcare sciences. Most of the students were hostel-based, so they often stored medicine in their rooms and bargained or sold these at lower costs to those who were unable to walk to a physician due to busy study schedules and unavailability of a pharmacy in a defined distance.

A study conducted by Alsous et al. (2018) aimed to highlight factors associated with self-medication in order to plan future interventions to minimize its risk. They concluded that self-medication is a usual practice in Jordan. People living in Jordan are frequently using names of various brands and are familiar with drugs appropriate for some common illnesses. Therefore, a very small proportion of people seek professional help to consult in a specific condition, thinking it to be a negligible illness [21]. The current study also noted that it was very rare that students were only using traditional medicines against coughing, flu, and diarrhea, and chest pain, while the most frequent medicinal categories were antimicrobial agents, anti-inflammatory, multivitamins, and for gynae purpose. A study from Sudan also highlighted that the self-medication was mainly performed in case of respiratory tract infections (38.1%) and cough (30.4%) [15].

Antibiotics were most commonly used to treat a sore throat, long-term fever, chest infections, and other upper respiratory infections. High consumption of antibiotics was found in most of students who were registered under the health sciences, as well as having a relationship with healthcare workers. They had good knowledge about the antibiotics and other medicines too. The data also revealed that self-dependence in preventing minor symptoms or conditions is another major reason of self-medication. Friends’ suggestions were also found to be an influential factor in self-medication in students.

According to Klemenc-Ketis et al. (2010), who conducted a study to determine the extent of self-medication among the healthcare as well as non-health care students, as well as the effect of various types of curricula on their attitude towards self-medication, the majority of students from either background had experienced some sort of self-medication during their student life. However, among healthcare students, seniors practiced self-medication more as compared to juniors. A web-based questionnaire, composed of three distinct sections, was used to assess the behavior of 1294 students. They found that non- healthcare students, on the other hand, took suggestions from their friends and colleagues in this regard [22].

The present study discovers the age bracket, i.e., 20–25 years old, who are highly exposed to self-medication. A high ratio of broad-spectrum antibiotics was consumed due to the fact that those from the narrow spectrum did not work as well; the broad-spectrum antibiotics aided the ill to recover early, even within few days of the initial infection or illness.

Ruiz et al. (2010) suggested a direct need of a proper monitoring system for the use of drugs and a partnership between healthcare professionals, patients, and pharmacists along with specific training and awareness programs about the risks of self-medication [23].

James et al. (2016) proposed a common tendency of using herbs, brews, medications when people felt unwell in Bahrain. However, expecting a different and more learned attitude from the students of medical background, they conducted a study to determine the knowledge and attitudes towards self-medication of first-year medical students. They were astonished to find that self-medication was practiced very commonly in a wrong manner. Namely, 44.8% of the subjects used self-medication in common conditions such as headache, flu, fever, cough, sore throat, and stomachache, while 81.3% of them used analgesics without knowing its side effects, appropriate dosage, and duration, thus displaying inadequate knowledge about self-medication [24].

## 4. Materials and Methods

### 4.1. Study Area and Selection of Study Participants

The study was conducted in selected departments of the different universities in the Hazara region, Khyber Pakhtunkhwa (KPK), Pakistan. The survey was conducted from July 2021 to October 2021. The Raosoft sample size calculator (Raosoft Inc., Seattle, WA, USA) was used to calculate the representative sample size. With an estimated 150,000 students in the Hazara region of KPK, it was computed using an expected 50% response rate, a 99% confidence interval, and a 5% margin of error. The calculated sample size was 636. However, the current survey has 1250 respondents, which is about twice the needed sample size, indicating that a convenience sample was used as the sampling strategy.

The survey questionnaire was designed in English and reviewed by a panel of professionals who gave comments on the various survey questions, which were subsequently revised based on their recommendations. A pilot study was undertaken with 20 participants who were excluded from the original assessment later to examine the clarity and comprehensibility of the survey content, and additional adjustments were made based on their feedback. The survey tool’s reliability was determined using the Cronbach’s alpha test of internal consistency. Cronbach’s alpha was 0.82, which was higher than the standard cut-off value of 0.70, so the test for this study showed that the survey tool was generally reliable.

All of the study participants were selected based on their willingness to answer the questions in the current study without any specific inclusion/exclusion criteria for gender or age. A series of interviews was arranged based on the study discipline, department, and the location of institution. All of the selected participants were divided into four major phases such as Phase-1 (all the undergraduate students), Phase-2 (Master (minor) students), Phase-3 (Master (major) students), and Phase-4 (Doctorate students). Before proceeding with the questionnaire distribution, the study participants were introduced to the objectives of current study and asked for their signed consent. Those study participants who were not willing to sign their consent were excluded from the study. This little introduction about the study decreases the selection bias of study participants associated with positive interest to very low or none. A soft copy of the questionnaire was also generated by Google forms for easy access for participants who preferred to answer the questions online (https://docs.google.com/forms/d/e/1FAIpQLSckkWMyJTybV7ktkO-EHRj3293JNtS5tD9dDLUYqhPmpvPzKQ/viewform?vc=0&c=0&w=1&flr=0).

It was also ensured that each response was provided once and not multiple times. In the Google form, the option to fill once only was enabled, while in case of physical forms, the patient names and identities were cross-checked later. The current study only included those students who were studying in universities, while students who were studying in secondary schools or colleges were excluded from the study.

### 4.2. Interview Calendar

The survey was arranged to conduct the interview at different universities based on the availability of the students during the study duration in Hazara Region of KPK, Pakistan. All students showed a positive interest towards the study and participated with a desire to learn something new as well as to know about the real facts. After the student agreed to participate in the current study, the questionnaire copy was provided to them, and they were asked to answer each question.

### 4.3. Sample Processing

The study included *n* = 1250 subjects from different universities of Hazara Region in KPK. A series of questionnaires was developed using Microsoft Word and Excel 2016. All the questionnaires were circulated among the students of different departments in various universities such as microbiology, zoology, chemistry, psychology, agriculture, medical lab technology, sociology, Pakistan study, food science, and business admiration.

### 4.4. Statistical Analysis

All the statistical analysis was carried out through SPSS 28. The study used Pearson correlation coefficient r = Σ(X − Mx) (Y − My)/(N − 1) SxSy, standard deviation, mean, variance, and probability for the desired data.

## 5. Conclusions

The most potent factors that were shown to influence self-medication in students were physician fee exemption, a convenient and cheap source of medication, and a lack of availability of pharmacies/hospitals nearby the hostels. Many indicated their physician and pharmacist did not provide any counseling about the risks of antibiotics and self-medication and that they usually searched for the medicine on social media platforms and purchased it blindly from pharmacies without any solid prescription. Such high self-medication practices increase the demand of healthcare professionals and campaigns about the disadvantages of self-medication aimed at populations living in rural and remote areas.

## Figures and Tables

**Figure 1 antibiotics-11-00842-f001:**
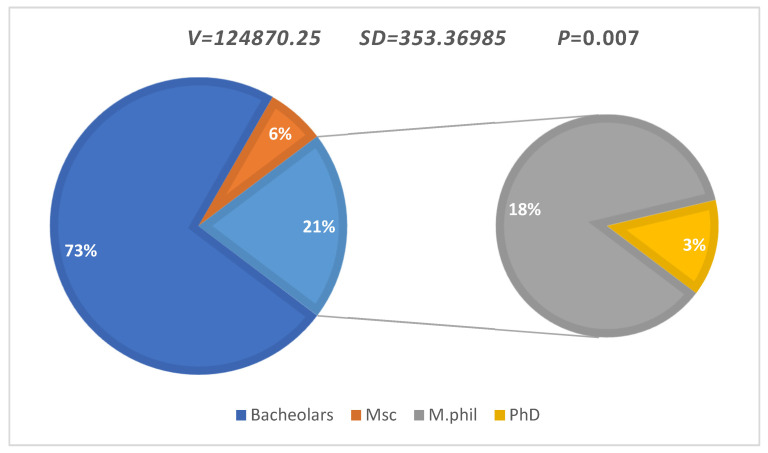
Discipline-wise students participated in survey.

**Figure 2 antibiotics-11-00842-f002:**
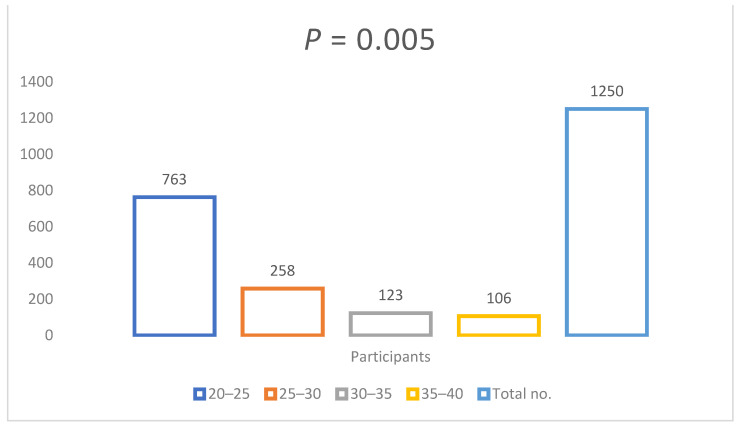
Age-wise students participated in survey.

**Figure 3 antibiotics-11-00842-f003:**
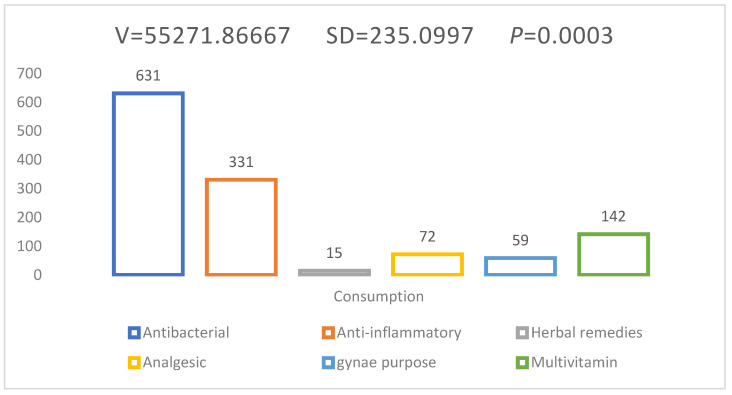
High antibiotics practices in students during routine life.

**Figure 4 antibiotics-11-00842-f004:**
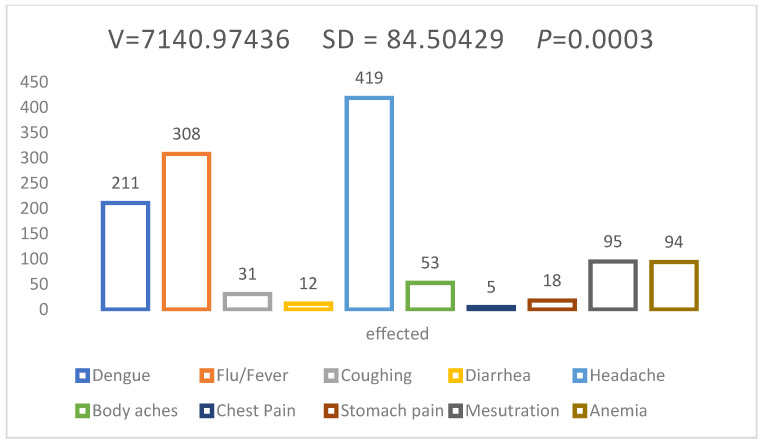
Self-medication practices against current infection/illnesses.

**Table 1 antibiotics-11-00842-t001:** Daily practices of the self-medication in students during routine life.

Serial No.	Questions	Keys	Scores	*p* Value
Q1	What is antibiotic?	Chemical used against the microbes	*n* = 205	0.002 *
		Reduce the disease or illness	*n* = 45	
		Inhibit or kill the micro-organism	*n* = 1005	
Q2	Do you currently use any of the following antibiotics or other medications?	Broad spectrum	*n* = 750	
		Narrow spectrum	*n* = 355	
		Others	*n* = 145	
Q3	Had you got any adverse reactions?	No	*n* = 991	
		Yes	*n* = 259	
Q4	Is self-medication better for human health?	Yes	*n* = 980	
		No	*n* = 208	
		Not noted	*n* = 62	
Q5	What is your concept for self-medication? Why you did choose it?	Reliable	*n* = 173	
		Exempt Physician fee	*n* = 241	
		Unable to walk at distance	*n* = 199	
		Self-satisfaction	*n* = 637	

* Significant association.

**Table 2 antibiotics-11-00842-t002:** Antibiotics knowledge in students during routine life.

Serial No.	Questions	Keys	Scores	Pearson Correlation Coefficient	(Mean) (Standard Deviation)	*p* Value
Q1	Do you know world is facing antibiotics resistance due to self-medication or misuse of antibiotics?	Yes	*n* = 570	R = 0.0311	M = 431.5	0.002 *
		No	*n* = 627			
		Not sure	*n* = 53			
Q2	Do you know high consumption of Antibiotics can damage your body organs?	Yes	*n* = 865			
		No	*n* = 138			
		Maybe	*n* = 247			
Q3	Do you know high consumption of Antibiotics can resistance against micro-organisms?	Yes	*n* = 782		SD = 1,615,917	
		No	*n* = 152			
		Maybe	*n* = 316			
Q4	As a self-medicated person, did a doctor or dentist ever talk to you about the dangers of antibiotics?	Yes	*n* = 421			
		No	*n* = 805			
		I did not see a doctor or dentist	*n* = 24			

* Significant association.

## Data Availability

Data related to the current study can be accesses upon a reasonable request to correponding authors.

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
