# Peer review of "Self-Medication as an Important Risk Factor for Antibiotic Resistance: A Multi-Institutional Survey among Students"

_antibiotics, 2022, doi:10.3390/antibiotics11070842_

Round 1
Reviewer 1 Report
This manuscript by Shah Zeb et al. describes the results of a multi-institutional survey of students.
This manuscript suffers from significant selection bias, and thus we cannot consider the results sufficiently robust.
The entire study needs to be rethought and the manuscript needs to be corrected :
Results: Line 155 "V and SD" is not needed.
According to Figure 1, we expect the results to be stratified by department of origin for the entire manuscript.
The structure of the manuscript needs to be corrected because the method considerations have been placed in the results section (and vice versa).
Part 2.6 is difficult to understand.
Parts 2.6.1.1 and 2.8.1.1 are not necessary.
Methods: how were the number of studies included determined?
How was selection bias associated with "positive interest" in the study considered?
How did the authors ensure that each response was provided once and not multiple times?
Section 4.5: What is the purpose of this section?
Section 4.6: What do the authors mean by "psychologically stable"?
Section 4.7: Because the authors considered exclusion criteria, they must provide a selection chart for the reader.
Author Response
Reviewer 1
Comments and Suggestions for Authors
This manuscript by Shah Zeb et al. describes the results of a multi-institutional survey of students. This manuscript suffers from significant selection bias, and thus we cannot consider the results sufficiently robust. The entire study needs to be rethought and the manuscript needs to be corrected:
Response: Dear reviewer, we would like to appreciate your comments in the current manuscript. We also would like to appreciate that, the comments from you and other reviewers have significantly improved the manuscript and it becomes better for the readers. Furthermore, we have revised the manuscript according to your comments and wherever necessary, we have answered you concerns about the flaws of manuscript. We hope that, after reading our responses to your comments you will agree to us.
Results: Line 155 "V and SD" is not needed.
Response: (Table 1) The columns about the V and SD values has been removed.
According to Figure 1, we expect the results to be stratified by departments of origin for the entire manuscript.
Response: Dear reviewer, we want to thank you for your valuable technical points on the current manuscript. The figure 1 has been removed from the revised manuscript and we have described the study participants in text. The reason to remove the figure 1 is, as we are unable to change all of the results as by department of origin.
The structure of the manuscript needs to be corrected because the method considerations have been placed in the results section (and vice versa).
Response: Dear reviewer, we would like to appreciate your valuable suggestion and efforts which improved the current manuscript. We have revised the manuscript and improved the structure of manuscript. Furthermore, the method consideration which were mistakenly placed in results section were moved to method section or removed.
Part 2.6 is difficult to understand.
Response: The section 2.6 has been revised.
Parts 2.6.1.1 and 2.8.1.1 are not necessary.
Response: Section 2.6.1.1 and 2.8.1.1 has been removed from the revised manuscript.
Methods: how was the number of studies included determined?
Response: (Line: 270-276) Dear reviewer, you might mean that how we determined the number of participants included for current study. We selected the Hazara region of Pakistan for the current random survey study. Before selecting the universities in the region, we have looked for the information about the departments in the concerned universities. Then based on the medical and non-medical departments we asked the concerned authorities to allow us to proceed with current study. Furthermore, the description about sample size calculation has been mentioned in the revised manuscript. Only those participants were included in the current study, who were willing to sign their awareness about the survey.
How was selection bias associated with "positive interest" in the study considered?
Response: (Line: 280-282) A sentence about selection bias and positive interest of the study participants has been added in the revised manuscript.
How did the authors ensure that each response was provided once and not multiple times?
Response: (Line: 300-302) The statement about ensuring that the survey was done once only by each of the participant has been added in the revised manuscript.
Section 4.5: What is the purpose of this section?
Response: The subsection 4.5 has been removed from the revised manuscript and have been added in section 4.2. In the previous version we added section 4.5 to highlight the more participants with age-scale because we added an age-scale section in results as a major part of the study so, in which we isolated most affected age such as 20–25 years with (61.0%) and it was significant age-scale of the study).
Section 4.6: What do the authors mean by "psychologically stable"?
Response: (Line: 286-288) The section 4.6 has been merged with section 4.2. The word “psychologically stable has been removed from the revised manuscript.
Section 4.7: Because the authors considered exclusion criteria, they must provide a selection chart for the reader.
Response: (Line: 303-304) The section 4.7 has been merged with section 4.2. Furthermore, we believe that this section doesn’t needs the chart anymore as we have mentioned each and every information in the revised manuscript.

Reviewer 2 Report
I have evaluated the manuscript (Antibiotics-1783430) titled “Self-medication as an important risk factor for Antibiotic resistance: A multi-institutional survey among students” by Ahmed and Co-workers, and the author has done a survey among students from August 2021 to October 2021 in Hazara Region of KPK, Pakistan on Self-medication which increased the risk factor for Antibiotic resistance. Excellent presentation of results in the manuscript and clearly describing the outcome. I found the document interesting for the readers and follow the scope of the journal Antibiotics.
I would like to recommend the article could be published in Antibiotics, with minor revision.
The authors could make the following minor changes.
1. Page 8, Line 257: Complete the incomplete sentence “Antibiotics were most commonly used to treat”.
2. A gender base survey could be useful here.
3. The author could have shown in the table the percentage of a particular drug (antibiotics or antimicrobials) used by students.
4. The author could include the questionnaire that was generated by Google form in the supplementary.
5. The author could include the following relevant references:
(a) Rather, Irfan A et al. “Self-medication and antibiotic resistance: Crisis, current challenges, and prevention.” Saudi journal of biological sciences vol. 24,4 (2017): 808-812. doi:10.1016/j.sjbs.2017.01.004
(b) Owusu-Ofori, A. K., Darko, E., Danquah, C. A., Agyarko-Poku, T., & Buabeng, K. O. (2021). Self-Medication and Antimicrobial Resistance: A Survey of Students Studying Healthcare Programmes at a Tertiary Institution in Ghana. Frontiers in public health, 9, 706290. https://doi.org/10.3389/fpubh.2021.706290
Author Response
Reviewer 2
Comments and Suggestions for Authors
I have evaluated the manuscript (Antibiotics-1783430) titled “Self-medication as an important risk factor for Antibiotic resistance: A multi-institutional survey among students” by Ahmed and Co-workers, and the author has done a survey among students from August 2021 to October 2021 in Hazara Region of KPK, Pakistan on Self-medication which increased the risk factor for Antibiotic resistance. Excellent presentation of results in the manuscript and clearly describing the outcome. I found the document interesting for the readers and follow the scope of the journal Antibiotics. I would like to recommend the article could be published in Antibiotics, with minor revision.
The authors could make the following minor changes.
- Page 8, Line 257: Complete the incomplete sentence “Antibiotics were most commonly used to treat”.
Response: (Line: 234-235) The sentence has been corrected.
- A gender base survey could be useful here.
Response: Dear reviewer, thank you for your valuable suggestion. Unfortunately, in the current manuscript we did not divide the study participants on gender basis. As at the time of study, we think that the it might not useful for the study site.
- The author could have shown in the table the percentage of a particular drug
Response: Dear reviewer, we would like to appreciate your valuable suggestion about to add the particular drug but unfortunately, in the current manuscript we did not work on which type of antibiotic the participants use. We will really appreciate, if you allow if you waive of the query as it is quite impossible to get back to the study participants and ask them about the antibiotic types.
- The author could include the questionnaire that was generated by Google form in the supplementary.
Response: Dear reviewer, thank you for your valuable suggestion. Instead of adding the questionnaire separately as supplementary file, we have added the web link of Google form in section 4.1. of the revised manuscript.
- The author could include the following relevant references:
(a) Rather, Irfan A et al. “Self-medication and antibiotic resistance: Crisis, current challenges, and prevention.” Saudi journal of biological sciences vol. 24,4 (2017): 808-812. doi:10.1016/j.sjbs.2017.01.004
(b) Owusu-Ofori, A. K., Darko, E., Danquah, C. A., Agyarko-Poku, T., & Buabeng, K. O. (2021). Self-Medication and Antimicrobial Resistance: A Survey of Students Studying Healthcare Programmes at a Tertiary Institution in Ghana. Frontiers in public health, 9, 706290. https://doi.org/10.3389/fpubh.2021.706290
Response: Dear reviewer, we would like to appreciate that you have provided us with the good articles, which we used for improving the introduction section. Furthermore, we have added these relevant citations as well.

Round 2
Reviewer 1 Report
The manuscript has been revised according to my previous comments.